# Effects of the Traditional Mediterranean Diet in Patients with Otitis Media with Effusion

**DOI:** 10.3390/nu13072181

**Published:** 2021-06-24

**Authors:** Fernando M. Calatayud-Sáez, Blanca Calatayud, Ana Calatayud

**Affiliations:** Child and Adolescent Clinic “La Palma”, C/Palma 17, bajo A, 13001 Ciudad Real, Spain; blanca.calatayud@gmail.com (B.C.); anacalatayud94@gmail.com (A.C.)

**Keywords:** otitis media with effusion, acute otitis media, rhino-sinusitis, Mediterranean diet, nutritional intervention

## Abstract

Introduction: Otitis media with effusion (OME) is common in pediatric primary care consultations. Its etiology is multifactorial, although it has been proven that inflammation factors mediate and that immunity is in a phase of relative immaturity. The objective of this study was to assess the effects of the Traditional Mediterranean Diet (TMD) modulating inflammation and immunity in patients diagnosed with OME. Materials and Methods: A analysis as a single-group pre-test/post-test was conducted on 40 girls and 40 boys between 18 months and 5 years old. Tympanometry normalization was the main test to control the benefit of diet. Clinical and therapeutic variables were studied through evaluation questionnaires, a quality test of the diet, as well as various anthropometric parameters. Results: At the end of one year, tympanometry had normalized in 85% of patients. The remaining 15% had normal audiometry and/or associated symptoms had decreased. Likewise, episodes of recurrent colds decreased from 5.96 ± 1.41 to 2.55 ± 0.37; bacterial complications of 3.09 ± 0.75 to 0.61 ± 0.06 and persistent nasal obstruction of 1.92 ± 0.27 to 0.26 ± 0.05. The degree of satisfaction of the families with the program was very high. Conclusions: The application of the Traditional Mediterranean Diet could have promising effects in the prevention and treatment of otitis media with effusion.

## 1. Introduction

Otitis media with effusion (OME), also called otitis media serosa, oto-tubaritis or otitis media with chronic exudate, is a chronic inflammation of the middle ear characterized by the presence of fluid buildup in the middle ear space in the absence of acute inflammation [1,2]. It is a very common disease in childhood, with 80% of children being considered to have experienced it at some point by the age of four. Among three year olds, its prevalence is estimated to range between 10% and 30%. Its most common symptom is mild hearing loss, which goes unnoticed by parents and is not easy to for paediatricians to detect in the absence of tympanometry devices [3,4]. Otoscopy can make an approximation to the diagnosis. However, otoscopy is not sensitive enough to detect many cases of OME [5]. It is controversial whether prolonged hearing loss in early childhood—a critical age for language acqisition—negatively influences a child’s educational development.

Although OME has a benign evolution, and a high percentage of spontaneous healing, treatment is generally aggressive, with a tendency to long-term use of drugs and to surgical intervention. The effectiveness of drug treatment, which includes the use of decongestants, antihistamines, oral and intra-nasal corticosteroids and antibiotics is limited, with few clinical benefits and an increase in side effects [6]. Self-insorwing techniques with insufficient results have also been described [7]. The American Academy of Paediatrics recommends monitoring these children; if the duration of the thundering is longer than three months and hearing loss is greater than 40 dB, the introduction of ventilation tubes is recommended [8]. Surgical treatment options include insertion of a tympanic drain, myringotomy and adenoidectomy. The benefits of surgical treatment are questioned by two systematic reviews, as hearing difficulties usually disappear spontaneously over time and levels of language and general development are equal in children with, or without, ventilation tubes [9,10,11]. We have found very few articles in the literature that link diet to the development of OME. According to the research we have been developing, a significant reduction in upper respiratory tract infections (URIs), acute otitis media (AOM) and persistent nasal obstruction, has been observed when children followed a high-quality diet [12,13]. The objective of this study was to assess the effects of the traditional Mediterranean diet (TMD) on the evolution of otitis media with effusion.

## 2. Materials and Methods

### 2.1. Study Design

The design correspond to a analysis as a single-group pre-test/post-test in which all patients aged 1 to 5 years consecutively diagnosed with OME. The study was conducted at a primary care pediatrics consultation, during the period from September 2010 to December 2018. All parents or guardians gave informed consent. The intervention focused on a food re-education based on the TMD through the use of the nutrition education program “Learning to eat from the Mediterranean”, which was used in previous studies [12,13]. This program consists of a series of visits with the nutritionist and the pediatrician, who proposed to assist the family. Visits are monthly for the first 4 months and bimonthly until the year is completed. The first visit evaluates the diet made by each child and his/her family, and changes in the usual diet are proposed making schemes, culinary recipes, example menus, etc. A baseline anthropometric assessment is also performed. Patients were monitored over the course of a year, valuing weight, stature, growth, clinical evolution, treatment needs, adherence to TMD and the degree of satisfaction of families (Figure 1). The study was approved by the Research Committee of the University General Hospital of Ciudad Real (Internal code: C-95, Act 03/2017).

### 2.2. Measuring Study Parameters

#### 2.2.1. Clinical Parameters

The main variable was the presence of OME. Otitis media with effusion was considered when the bilateral exudate or effusion persisted for more than 3 months, or more than 6 if it is unilateral [14,15]. The presence of trans-tympanic fluid was assessed by otoscopy and with the help of a portable tympanometer (MicroTymp 3^®^). The following secondary variables were considered: Number of upper respiratory tract infections (URI), number of episodes of OMA and rhino-sinusitis (RS), emergency assistance, number of symptomatic drugs and prescribed antibiotics, all of which are valued per person per year. A basic otorino-laryngological scan was done, which included rhinoscopy, pharyngoscopy, otoscopy, tympanometry, audiometry in collaborating children with a portable audioscope (Audioscope^®^), intentional face appreciation (adenoid facies) and cervical adenopathies and finally a clinical assessment was made of the degree of persistent nasal obstruction (PNO): mild-1, moderate-2, or severe-3. An episode of URI was defined by two or more of the following criteria: Fever greater than 38 °C measured with a tympanic thermometer, nasal congestion or oral breathing, nasal discharge, odynophagia and cough. AOM was defined according to the criteria of the Guide of the American Association of Pediatrics [16] are: 1. Acute presentation; 2. Presence of exudate in the middle ear cavity demonstrated by tympanic bulge, pathological or otorrhea pneumatoscopy; 3. Inflammatory signs and symptoms such as otalgia or evident redness of the eardrum. When the diagnosis of AOM was confirmed, conservative or expectant treatment was followed in mild cases and antibiotics were administered in cases of increased risk, such as in children under 2 years of age, bilateral AOMs, and those with increased general affectation. It was defined as rhino-sinusitis to the persistence of daytime or rhinosherhorrhone cough for more than ten days, with no apparent improvement, in the context of an upper respiratory infection [17]. Persistent nasal obstruction was defined as difficulty breathing properly nasally, with associated respiratory symptoms, such as oral breathing, snoring, difficult breathing in sleep, stopping respiratory sleep (apnea), agitated sleep, neck stresses to sleep, drowsiness or feeling of not having rested properly, adenoid facies and swallowing difficulties [18].

#### 2.2.2. Clinical and Therapeutic Evaluation Rate Performed on Parents or Guardians

To assess the clinical evolution of the patients, a questionnaire was designed, addressed to parents or guardians, in which the symptoms related to OME were evaluated and their subsequent evolution, the intensity of clinical tables, the needs of difficulties with the diet and, the degree of satisfaction with the intervention. To each question in the questionnaire, one can answer the improvement observed with: 3: much, 2: quite, 1: something, 0: nothing. 

Ten questions referred to the clinic and treatment in the previous four weeks and a maximum of 30 (good control) to a minimum of 0 (poor control) was scored. 

A patient was considered to be poorly controlled when the total score was equal to or less than 20. (Table 1)

#### 2.2.3. Weight-Statutory Evolution Parameters

Anthropometric data were collected, such as weight, height, skin folds, perimeters of the arm, abdomen and waist according to standardized procedures and with them the body mass index, lean mass and body fat mass were calculated [19].

#### 2.2.4. Parameters of Accession to the TMD

Adherence to the diet was assessed using the Kidmed test, and the TMD test, which was discussed in previous works [12,13,19]. The KidMed test is one of the most prestigious for evaluating the quality of children’s nutritional intake based on the TMD. It consists of a questionnaire of 16 questions that must be answered affirmatively/negatively (yes/no). Affirmative answers to questions that represent a negative connotation in relation to the Mediterranean diet (there are four) are worth −1 point, and affirmative answers to questions that represent a positive aspect in relation to the Mediterranean diet (there are 12) are worth +1 point. Negative answers do not score. Therefore, this index can range from 0 (minimum adherence) to 12 (maximum adherence).

In order to measure the newly proposed points, we developed a complementary test (the Traditional Mediterranean Diet Test or Test-TMD) with the same structure, to which we have added nutritional and behavioural questions that—in our opinion—are not reflected in the KidMed test. This test consists of 20 questions that must be answered affirmatively/negatively. Unlike the KidMed test, in Test-TMD, all the questions are positive. They are therefore scored with one point for each affirmative answer, and the results can range between 0 and 20 points. A test score below or equal to seven points is considered ‘poor quality’, a score between eight and 14 points is considered as ‘need to improve’ and scores above 15 points are considered as ‘optimal traditional Mediterranean diet’. At each visit, we evaluate the nutritional tests and together with the patients and their parents, we analyse any difficulties that may have arisen and examine how we could modify behaviour to obtain the best results. Both questionnaires allow the KidMed index and the TMD index to be calculated. According to scores obtained in the KidMed questionnaire, three degrees of quality of the Mediterranean diet can be obtained: a) ‘good’ or ‘optimal’, when the score is equal to or greater than eight; b) ‘average’ or ‘need to improve diet or nutritional habits’, when the score is between four and seven, inclusive; and c) ‘poorly adapted’ or ‘low-quality diet’, when the score is equal to or less than three. According to the scores obtained in the TMD index, three grades are obtained: low quality ≤7, moderate quality 8–14, optimal quality >14.

The traditional Mediterranean diet is characterized by a high content of fresh, raw, perishable and seasonal foods, rich in vegetable fiber, minerals, vitamins, enzymes and anti-oxidants; abundant fruits, vegetables, legumes, and whole grains, one of whose characteristics is its low-moderate glycemic index; sufficient polyunsaturated fats from crude oils, nuts, seeds and fish; low protein and saturated fat content of animal origin and, a low use of precooked and industrial foods. This means, in daily practice, the limitation of products such as white bread, industrial pastries, cow’s milk, red and processed meats, sugary industrial beverages and precooked fast food [20]. The TMD is based on the Decalogue that the Foundation of the Mediterranean diet proposes to us through its website (Table 2) [21].

This has been proclaimed cultural heritage and intangible heritage of humanity by Unesco [22]. In Table 3, we expose the differences between TMD and the diet promoted by “Western civilization”.

#### 2.2.5. Sample Size and Statistical Analysis 

To calculate the sample size, a significance level of 0.05 and a power of 80% was used, assuming an OME decrease of 50% and a standard deviation of 3.5 units, adjusting for a 25% loss, which resulted in a sample size of 80 patients. For the analysis of the results, the statistical package SPSS 15.0 was used. A descriptive analysis was carried out with statistics of central tendency and dispersion for quantitative variables, absolute and relative frequencies for qualitative variables. The comparison of the results of the different variables before and after the intervention was carried out by means of the Student’s t-test for paired data when the variables followed a normal distribution, or by the Wilcoxon test when they did not adjust to normal, after checking with the Shapiro-Wilk test.

## 3. Results

Participation was proposed in a program called ‘Learning to eat from the Mediterranean’. The families of 93 patients met the OME inclusion criteria. Seven refused to participate. Form the 86 patients included, six left the program after the first sessions. Two weree due to social or personal difficulties in implementing the diet, one was due to disagreement with limitations of certain foods and three due to surgical interventions indicated by the OR service and not coordinated with our team. The study was thus completed with a total of 80 patients (40 girls and 40 boys) with an average age of 3.1 years. All patients included in the study were evaluated at four and 12 months after the initial visit. The results obtained were similar in both sexes, and are thus, collated together (Table 4).

During the treatment year, there was a normalization of tympanometry and therefore a reduction in the number of patients affected by OME to 85% of total participants, while the remaining 15% had improved hearing and/or associated symptoms. The number of URI and bacterial complications also decreased (Table 5). 

The level of household satisfaction was high, as shown in the questionnaire on observed improvements (Table 1). Anthropometric variables before, at four months and after intervention, are set out in Table 6. There was an adequate, statistically significant increase in determining parameters of growth and development, such as size and lean mass. The average weight increase in the year prior to the study was 2.01 kg, compared to 2.45 kg post-intervention, and the average size increase was 6.6 cm, compared with 7.30 cm post-intervention. Body mass index (BMI) decreased. The lean mass area of the arm increased, while the area of fat mass decreased.

At the end of the program, patients’ dietary habits had also improved in the sample as a whole, with an increase observed in the number of patients consuming fruits, vegetables, fish, whole grains and fermented dairy. Additionally, the percentage of patients who did not eat breakfast or who had industrial breakfast pastries decreased, as did the proportion of those who consumed treats on a daily basis. The mean value of the KidMed index at the beginning of the program was 7.09 ± 1.82 points. 52.29% of the patients obtained a qualification according to the KidMed test of “need to improve” and 47.71% obtained the qualification of optimal diet. At the end of the study, 98.6% of the children obtained optimal levels with a mean of 9.21 ± 1.29 points, mean difference of 2.12 ± 0.11 (95% CI: 1.90–2.30 *p* < 0.01). According to this data, the average value of the KIDMED index evolved from a score considered medium-high at the beginning of the program to an optimal value at the end of the program (Table 7).

At the beginning of the study, the mean value of the DMT-Test was 6.91 ± 1.98, qualifying as a poor quality diet. 82.3% of the sample obtained a score below 8 points (poor quality diet) and 17.7% obtained a score between 8 and 14 points (need for improvement). At the end of the study, the mean score was 16.3 ± 1.90 points, qualifying as an Optimal Traditional Mediterranean Diet. (Figure 2). The TMD test evolved from levels considered to be low quality to optimal levels (Table 8 and Figure 2).

## 4. Discussion

In view of these results, we suggest that the Traditional Mediterranean Diet could help in the prevention and control of otitis media with effusion. We have been able to verify that children who have followed our dietary recommendations have improved the inflammatory response and the defensive capacity against the usual infectious diseases. The study shows that a high percentage of patients with OME evolved satisfactorily with the use of TMD; tympanometry normalized in 85% of children. A decrease in the number of recurrent respiratory tract infections and their most frequent complications was also observed. The degree of nasal obstruction (PNO) also decreased. Although flat tympanometry persisted, in the remaining 15% of OME, their hearing and/or their symptoms improved. There was less use of symptomatic drugs, anti-inflammatory drugs and antibiotics. Emergency visits decreased and the degree of family satisfaction was high (Table 1). The number of URIs with bacterial complications decreased by 65% (3.12 from the previous year compared to 1.08 from the year of intervention). The average URI, compared to the previous year, was down 59%; 62% of patients had no bacterial complications during the nutritional intervention period, 29% had only one in the whole year and 10% had two, compared to the three episodes they had on average in the previous year. Children with PNOs went from a mild-moderate to nothing-mild intensity profile. Emergency assistance also decreased by 70%. Antibiotic use decreased by 84.3%, and the use of symptomatic drugs decreased by 57.5%. The majority of patients did not require surgical intervention, and clinical evolution suggests that it will no longer be necessary. This better way to defend against prevalent childhood diseases, together with the best anti-inflammatory response, could be the cause of the progressive decline in PNO in our pediatric population.

The main difficulty was the fulfillment of the diet, as they were proposed to make a homemade diet, family and fresh products that must be prepared and not always the parents had time and dedication to do it properly. The dietitian-nutritionist follow-up contest was essential to ensure compliance. By the end of the program, the dietary habits of the patients had improved in the sample as a whole: An increase in the number of patients consuming fruits, vegetables, nuts, whole grains and fermented dairy products was observed. In general, the consumption of proteins of animal origin was reduced considerably, especially cow’s milk, red meats and meat products. The consumption of processed foods also decreased, especially industrial pastries. The patients showed satisfactory predicted growth rates. Their weight, height and BMI percentile evolved as expected. A positive result was the slight decrease in BMI and fat mass levels and a small increase in height and lean body mass. We have subsequently followed up the children who participated in the study and have not had a recurrence of OME. Some families relaxed the fundamentals of the TMD over time, which led to the development of other inflammatory diseases. This, in turn, required reimplementation of the TMD [12,13].

It is important to note that during the time that the incorporation of patients to the study lasted, we extended the application of TMD to the entire pediatric population (siblings, relatives, patients with other recurrent pathologies, and infants under two years of age). This led to a progressive decrease in patients diagnosed with OME, so the achievement of the sample size was delayed. All this has resulted in a decrease in patients diagnosed with OME. It has come become a rare disease in children of our pediatric quota who follow TMD [23]. The growing interest in the Mediterranean diet is based on its role in inflammatory diseases [24]. Several clinical and epidemiological studies, as well as experimental studies show that the consumption of the DMT reduces the incidence of certain pathologies related to oxidative stress, chronic inflammation and the immune system, such as cancer, atherosclerosis or cardiovascular disease [25]. There is evidence that diet and individual nutrients can influence systemic markers of immune function and inflammation [26]. However, there is no data on its direct action in the pediatric patient. Some data suggest that the follow-up of a diet with an excess of refined flours and processed foods of animal origin, together with an infrequent consumption of fruits and vegetables, is associated with high inflammatory markers [27]. In studies conducted on mucoid samples of patients with OME, there was a global tendency to increase local pro-inflammatory mediators [28,29]. An increase in the markers of oxidative stress in the effusion of children with OME has also been demonstrated, which would lead to a pro-inflammatory and hyper-reactivity state of the mucosa against infectious agents [30,31,32,33]. The excess “antigenic load” inherent in the Western diet of today, which has multiplied available foodstuffs by the thousand may misadjust our immune system, making it weaker and notably hyperplasic. Children with OMAR show immaturity in antigen presenting cells with a suboptimal response of T cells and B [34] memory. The absence of Toll-2 receptors (TLR2) can lead to prolonged inflammation of the middle ear. TLR2 is essential for the timely resolution of inflammation since it has been proven to promote macrophage recruitment and bacterial clearance in the mouse [35]. The pro-inflammatory actions of PAF (platelec-activating factor) can be favorably modulated with DMT and regulate its metabolism [36].

In recent years, patients diagnosed with food intolerance to cow’s milk proteins and other non-IgE-mediated foods, which cause inflammation in the digestive and respiratory mucous membranes with various symptoms, are increasing. The current treatment of these entities (food intolerances not mediated by IgE, eosinophilic esophagitis, sensitivity to non-celiac wheat, etc.) involves the elimination of the proteins involved [37,38,39]. We believe that sensitization to these proteins may be at the base of the inflammation of the mucosa of the middle ear and for this reason our patients could have responded adequately to their elimination in the diet. Data from several randomized DM-based clinic trials, have demonstrated a beneficial effect in the primary and secondary prevention of disease. The exact mechanism by which an increased adherence to the TMD exerts its favorable effects is not known, although they have been shown to protect against oxidative stress and inflammation [40].

On the other hand, the microbiota depends to a great extent on the food we eat, and we consider that this issue should be studied in depth in future studies. The follow-up of the DMT is associated with a more beneficial microbiota profile for health, with a higher production of short chain fatty acids, the presence of Prevotella and some Firmicutes capable of degrading fiber [41]. Diet has a rapid effect on gut microbiota composition, which promotes the growth of certain bacterial groups over others, as well as changes in intestinal pH, intestinal permeability, bacterial metabolites, and thus, inflammation [42]. The available evidence suggests that gut microbiota of subjects that follow the TMD is significantly different from subjects that follow a Western diet model. The latter shows an increased gut permeability, which is responsible for metabolic endotoxemia. For this reason, we can speculate that the gut microbiota of the subjects following the TMD is able to prevent the onset of chronic non-communicable diseases [43]. The nasal microbial composition in children with OME is less diverse and with a greater abundance of pathogens than their peers without respiratory infections [44]. The changes in the levels of citocines, may indicate bacterial pathogen as one of the causes of OME [45]. The presence of Alloiococcus otitidis has been considered as a precipitating factor of OME [46]. Likewise, the importance of “bacterial biofilm” and its association with persistent otic disease has been observed [47,48,49]. The local administration of probiotic bacteria seems to have the ability to inhibit the growth of otopathogens [50]. It is interesting to consider the relationship of DMT with the nasopharyngeal microbiome and the role in the treatment or prevention of OME. The results of this intervention study would point to a possible mechanism of action.

One of the characteristics that every research study should have is that it is easily reproducible, using small groups, and with a little economic cost. The work presented here is easy to reproduce in any primary care pediatric consultation, but it is not easy to perform, due to the lack of nutritionists and the lack of effective monitoring of the diet. Our study has limitations, and particularly the lack of a control group, which would have allowed us to compare the results. We could not perform the study with a control group given that most of our paediatric space adhered to the Mediterranean diet and did not seem ethical to promote a standard western-type diet in a control group. Our hypothesis is that the standard diet proposed by “Western civilization” is the origin of alterations in the inflammatory and immune mechanisms and therefore the cause of most childhood diseases. Although with age comes a slow spontaneous tendency to resolve OME, such a rapid disappearance of symptoms could not be expected. A notable decrease was found in the number of children who required pharmacological and surgical treatment. Therefore, we deduce that the nutritional intervention was beneficial. Pre-test/post-test studies, such as ours are prospective and provide a moderate level of evidence. Most importantly, the results of this research could support further studies on the influence of the Mediterranean diet on this, and other chronic and recurrent inflammatory diseases. 

It would have been very interesting to perform analyzes that measured the response of the immune system, inflammatory markers and data on the modification of the microbiota when making the nutritional change. The present study is only part of a general project that we are carrying out, which covers most of the recurrent diseases of childhood. Most of our patients have been consecutively included in the program “Learning to eat from the Mediterranean” and we have verified how the prevalence of OME and other recurrent diseases has decreased considerably. It should not go unnoticed, the change of “model of medicine” that these research studies entail. It is no longer about remedying a disease with external drugs, outside the defensive system or limiting surgical interventions, but the therapeutic proposal is based on providing the body with everything it needs to solve their needs and eliminate that for which it is not ready. We can conclude by saying that the application of the Traditional Mediterranean Diet could have promising effects in the prevention and treatment of otitis media with effusion, with the normalization of tympanometry and hearing loss, with a notable decrease in associated inflammatory diseases, use of drugs and placement of grommets.

## Figures and Tables

**Figure 1 nutrients-13-02181-f001:**
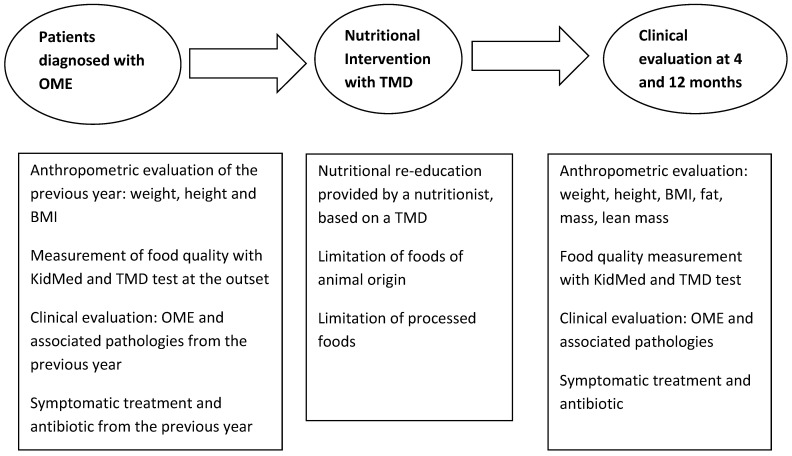
Study Design diagram.

**Figure 2 nutrients-13-02181-f002:**
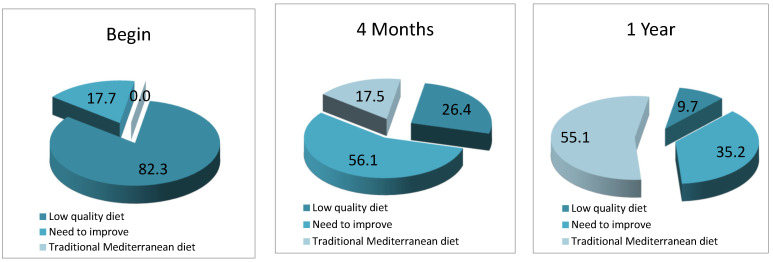
Evolution of quality of diet, measured using TMD Test.

**Table 1 nutrients-13-02181-t001:** Clinical and therapeutic evaluation index. Responses from parents or guardians regarding the improvement observed: 3: much, 2: quite, 1: something, 0: nothing.

Clinical and Therapeutic Evaluation Index	4 Month	1 Year
Has the number of highways colds decreased?	2.29	2.55
Have you noticed less intensity in the infectious processes?	2.70	2.95
Has the nasal obstruction decreased, breathe better?	2.96	2.95
Have complications decreased?	2.91	2.95
Do you hear better?	2.36	2.95
Have you noticed the least use of antibiotics?	2.50	2.95
Have you noticed the least use of symptomatic medications?	2.95	2.95
Has there been good diet tolerance on the part of the patient?	2.95	2.95
Has food quality improved?	2.96	2.96
Are you satisfied with the results?	2.50	2.96

**Table 2 nutrients-13-02181-t002:** Mediterranean diet. Ten basic recommendations.

Always Adpatinf the Diet to the the Child’s Needs (According to Age)
Use olive oil as your main source of added fat. Eat plenty of fruits, vegetables, legumes and nuts. Bread and other grain products (pasta, rice, and whole grains) should be a part of your everyday diet. Food that has undergone minimal processing, fresh and locally produced food is best. Consume dairy products on a daily basis, mainly yogurt and cheese. Red meat should be consumed in moderation and if possible as a part of stews and other recipes. Consume fish abundantly and eggs in moderation. Fresh fruit should be your everyday dessert and, sweets, cakes and dairy desserts should be consumed only on occasion. Water is the beverage par excellence in the Mediterranean Diet. Be physically active every day, since it is just as important as eating well.

**Table 3 nutrients-13-02181-t003:** Differences between the Traditional Mediterranean Diet and the “Western civilisation” Diet.

Traditional Mediterranean Diet	Western Civilisation Diet
Breastfeeding	Adapted milk
Varied, seasonal fruit	Baby food jars and canned fruits
Vegetables and leafy vegetables	Baby food jars and canned vegetables and leafy vegetables
Pulses and non-processed nuts	Canned pulses and dried, fried or salted nuts
Minimally processed and fermented whole grains	Refined, processed cereals with industrial fermenting agents
Fermented milk, principally goat’s and sheep’s	Whole, processed milks, mainly from cows
Occasional lean meat, in small quantities	High consumption of red, processed meats
Minimally processed, perishable, fresh and local foods	Nonperishable processed and ultra-processed foods
Limits on products with added chemicals	Presence of chemical agents and enzyme disrupters

**Table 4 nutrients-13-02181-t004:** Sample characteristics. Average age 3.1 years.

	Boys (*n* = 40)	Girls (*n* = 40)
Weight (kg)	13.16 ± 2.35	15.00 ± 2.21
Height (m)	0.90 ± 0.06	0.95 ± 0.07
BMI (kg/m^2^)	15.97 ± 1.46	16.42 ± 1.19
Fat mass (%)	15.60 ± 2.77	14.89 ± 1.95
Lean mass (%)	11.05 ± 1.78	12.76 ± 2.17

**Table 5 nutrients-13-02181-t005:** Evolution during the previous year and during the year of treatment.

	Previous Year to Treatment	One Year of Treatment	*p*
Otitis Media with effusion (OME)	100%	15%	0.001
Degree of involvement of otitis media with effusión (OME). 0: mild, 1: moderate, 2: intense	1.92± 0.27	0.26± 0.05	0.001
Number of upper respiratory tract infections (URI) per child and year	6.01± 1.41	2.52± 0.37	0.001
Bacterial complications per child and year (Acute otitis media and rhino-sinusitis)	3.12± 0.75	1.08± 0.06	0.001
Number of emergency treatments per child and year	1.90± 0.76	0.33± 0.04	0.001
Antibiotics per child and year	3.71 ± 0.37	0.58 ± 0.08	0.001
Symptomatic treatment per child and year	7.13 ± 0.96	3.03 ± 0.39	0.001

**Table 6 nutrients-13-02181-t006:** Anthropometric assessment at the start, after four months and after one year.

	At the Start of Treatment	4 Months of Treatment	Year of Treatment	*p*
BMI (body mass index)	16.19 ± 1.41	15.74 ± 1.03	15.45 ± 1.34	0.11
Fat mass (%)	15.23 ± 2.69	15.03 ± 2.49	14.76 ± 2.29	0.27
Lean mass (%)	11.79 ± 2.29	12.44 ± 1.98	13.82 ± 2.08	0.03

**Table 7 nutrients-13-02181-t007:** KidMed Test (percentage).

	At the Start	After 4 Months	After One Year
1 piece of fruit per day	80	98	100
1 + piece of fruit per day	29	74	88
1 vegetable per day	91	97	100
Vegetables more than once per day	17	42	71
Regularly eats fresh fish (2–3 times/week)	88	98	98
Visits fast food rest. once or more per week	28	5	2
Legumes 1–2 times/week	92	98	98
Pasta and rice every week	98	100	100
Cereal or deriv. for breakfast	71	98	98
Regularly eats dried fruit and nuts	20	28	31
Olive oil used at home	100	100	100
No breakfast	6	3	2
Dairy at breakfast	83	95	98
Factory-baked goods for breakfast	34	8	3
Two yoghurts or 40 g cheese/day	100	100	98
Sweets and snacks every day	22	3	2

**Table 8 nutrients-13-02181-t008:** TMD Test (%).

	Start	4 Months	Year
Minimum 2 pieces of fruit every day.	42.5	76.3	93.8
Fresh vegetables at every meal, as a first course or as part of the main course.	38.1	53.6	79.1
Limited sugar intake (sweetened breakfast cereal, sweetened yoghurts or milkshakes, cakes, soft drinks, sugary biscuits, sweets, ice-cream, etc).	6.3	72.5	76.2
Sporadic use of potatoes (1–2 times/week) and preferably not fried.	23.2	76.2	84.9
Legumes twice or more per week, not always with meat.	35.3	83.3	81.3
Regular intake of white fish, oily fish and seafood (1–3 times/week).	70.1	73.5	93.1
Preferably eats whole grains (whole wheat pasta, brown rice, brown bread, etc., limiting the intake of refined flour such as white bread to less than 40 g per day).	15.5	58.9	81.9
Intake of seasonal, natural, fresh food.	25.3	72.4	84.4
Moderate to low intake of dairy products: Preferably in the form of natural yoghurt and goat’s or sheep’s cheese.	14.6	66.5	86.4
Only lean processed meats, less than twice per week.	16.9	63.3	83.4
Preferably white meat, less than 3 times per week (lean).	25.5	73.4	87.8
30–50% of the daily intake consists of raw food (fruit, vegetables, virgin extra olive oil, freshly-squeezed fruit juice, nuts, etc.) and semi-raw food (green vegetables).	4.9	29.8	63.7
Frequent intake of broths, soups, natural smoothies and water.	43.3	68.9	68.9
Intake of fats mainly from virgin extra olive oil and raw nuts. Avoiding low quality industrial fats.	36.4	78.9	91.2
Good quality breakfast and mid-morning meal, without processed foods.	27.3	63.1	79.1
No snacking between meals and a reasonable portion size at meals.	28.7	65.4	88.2
Adapts to the food made at home (family) and alternatives not offered.	38.2	65.5	86.4
Limits intake of additives, avoiding “junk” food (<1/week)	75.6	83.6	82.7
Regular physical exercise (running, playing, walking, climbing, etc) or sport.	68.2	73.3	82.3
Mealtimes together, avoiding the television or other technology.	67.9	78.7	96.4

## Data Availability

Not Applicable.

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
