# Peer review of "Effects of the Traditional Mediterranean Diet in Patients with Otitis Media with Effusion"

_nutrients, 2021, doi:10.3390/nu13072181_

Round 1
Reviewer 1 Report
Very interesting paper, but as the authors suggested “the work is easy to reproduce in any primary care pediatric consultation, but it is not easy to perform, due to the lack of nutritionists and the lack of effective monitoring of the diet performed”. I think that also the involvement of the family had got a key role in the success but it would be difficult to maintain the good habits if not stimulated by the nutritionist. Would be interested to re-analyse the children after another year.
Revision point
Line 131:
How the BMI was calculated? For children is not corrected to calculated as weight (kg)/height (m)2 .but body mass index percentiles are needed. See the Centers for Disease and Control prevention charts available at http://www.cdc.gov/nchs/. Moreover, weight class must be defined according to the International Obesity Task Force classification [Cole TJ, Bellizzi MC, Flegal KM, Dietz WH. Establishing a standard definition for child overweight and obesity worldwide: international survey. Brit Med J. 2000;320:1240–3]. Revision of table 4 and 6 is needed.
Table 5 an 6: use period not comma for P
Line 190: a revision of BMI variation is needed, see first comment
Tabel 8: “Moderate to low intake of dairy produce:” Did you means dairy products?
Author Response
Consulte el archivo adjunto

Reviewer 2 Report
The manuscript by Calatayud-Saez et al aims to investigate the effect of a “traditional Mediterranean Diet’ on the condition otitis media with effusion. This is an intervention with a pre-test and post-test design in a pediatric population (1-5 years). While the fundamental study questions is interesting and may warrant investigation there are several concerns with the manuscript including the writing, study design, and interpretation. A major consideration is the suitability of this intervention in such a young pediatric population.
Mostly notably the manuscript contains numerous grammatical errors making critical sections such as the introduction and results difficult to interpret at times. In its current form, the writing is not suitable for publication.
Additionally, the rationale is not clear for why this intervention is suitable for such a young pediatric population (children as young as 12 months). There are major concerns in regards to applying this intervention to such a young pediatric population during critical periods of growth. Infants and toddlers often have a very narrow range of food intake and dietary modifications should be overseen by a physician. Not all components of the TMD can be adhered to by a 12 month old. It is not clear how this dietary intervention was applied to the youngest of children in an ethical manner and how adherence was measured. In addition, the quantity of foods eaten should be considered. Can an infant/toddler eat enough of TMD foods to elicit a biological effect? There are no details on how adherence was measured. The measures of adherence presented by the authors are likely confounded by growth and development.
The study pre-test post-test design utilized in this study is not ideal and results cannot be interpreted as a RCT with a control group. This approach does not demonstrate the effect of a treatment. This analytic approach suffers from a lack of internal and external validity. Importantly, this approach commonly suffers from regression to the mean, bias due to maturation of the participants (highly relevant in this sample of growing children), and cannot account for other medical events that may affect measurements. If the authors have three time points of assessment an ANCOVA should be used for the analysis. The authors assess many outcome measures with no correction for multiple testing. Measures of growth should be stratified by developmental age.
The discussion does not appropriately compare to prior literature and instead focuses on speculation about the gut microbiome which was not measured in this study. Results are interpreted with causality when at best they are hypothesis generating.
Reviewer 3 Report
The objective of this study was to assess the effects of the Traditional Mediterranean Diet modulating inflammation and immunity in patients diagnosed with otitis media with effusion, with the most common symptom mild lytebypass hearing loss, which goes unnoticed by parents and not easy to detect by the pediatrician in the absence of tympanometry devices.
The problem is that the effectiveness of drug treatment is poor, and includes the use of decongestants, antihistamines, oral and intra-nasal corticoster- oids and antibiotics for which little clinical benefits was found, but of an increase in side effects.
In this work, it turns out that the Traditional Mediterranean Diet can help prevent and also control otitis media with effusion.
For all the above reasons, the in-depth study of the subject in combination with the worthwhile team and the suitable methodology, make me consider the submitted manuscript as an Excellent one.
However, there are many publications in the literature in which Traditional Mediterranean Diet is related with the most powerful lipoid factor of thrombosis and inflammation, the Platelet Activating Factor (PAF). Inhibitors of this factor have been found in Mediterranean foods and are an explanation of the protective effect of the Mediterranean diet, as these PAF inhibitors inhibit inflammation.
The work would need to be strengthened by adding bibliographic references that includes all the relative knowledge and at least the article
Mediterranean diet and platelet-activating factor; a systematic review. Clinical Biochemistry, 60, Pages 1-10, 2018,
Nomikos, T., Fragopoulou, E., Antonopoulou, S., Panagiotakos, D.B
should be mentioned in the submitted manuscript.
Round 2
Reviewer 2 Report
Grammatical errors in the manuscript have been fixed. However, there are still major flaws with this study that were not fully addressed.
Study Design:
- As stated by the authors (response #2) and from reading prior publications this work appears to be from a larger study evaluating TMD and “usual inflammatory and recurrent diseases of childhood.” OME is the primary outcome of this paper the primary endpoint and secondary endpoints of this overall study are unclear. This context must be clearly provided for a proper interpretation of results. It is unclear how this study fits into the larger parent study and if the enrollment was different for each outcome. If the assessment of OME is actually a secondary outcome in a sub-group of the larger study this needs to be clearly stated. Importantly, all methods should be written with enough detail to serve as a standalone description for this manuscript.
- A diagram of the study design and specific time point of data collection would be helpful to the readers
- Study design line 63: the authors should refer to their analysis as a one-group pretest-posttest design which is how it is referred to in the scientific literature.
Mediterranean Diet:
Points of my initial comments regarding the Mediterranean diet have not been addressed.
- Particularly, it is not clear how this dietary protocol was adapted to different developmental groups? For example, were children without fully developed teeth advised to eat nuts? If not were they still penalized?
- How was the quantity of food eaten addressed? Children would be progressively eating more solid foods.
- Is this prior work (by external groups) suggesting a Mediterranean diet has anti-inflammatory properties in this population?
There are still major concerns about the use and reporting of the KidMed/TMD
- It is not clear why the KidMed test and the TMD test are used. The rationale should be stated.
- The KidMed test/Kidmed Index is not validated in children <2 years. Some criteria are not applicable to children <2 years.
- Exactly how the KidMed test was administered should be stated. Who administered this test to the children?
- The TMD does not appear to be validated in any population. This is important considering the subjective nature of some criteria (eg. “good quality breakfast”). Its utility in a pediatric population is unclear.
- The scoring criteria for the TMD is not stated. How is each component scored? What is considered “low quality”, “need to improve”, or TMD”?
- Adherence to the Med diet by the KidMed Index requires enough description to serve as a standalone in this manuscript.
- A reference to the original manuscript on the construction of the KidMed test should be included – references to the author's prior work which does not adequately describe the index are not needed in this section
- Scoring criteria, including negative and positive scoring by component, should be stated. Table 7 displays the KidTest but the KidIndex which is presumably used to measure adherence (line 186), is not shown.
- Overall the writing about methods pertaining to the Mediterranean diet just includes lengthy descriptions of the diet itself and lacks detail on the actual assessment. An adequate description is needed.
Statistical analysis, results, and limitations
- The statistical analysis section requires more details of the analysis
- Authors must specify the primary outcome and list all secondary outcomes. For each outcome, it should be stated if it was continuous, ordinal, or binary.
- Detail on statistical tests for binary and ordinal outcomes is needed.
- Is the power calculation post hoc?
- Correction for multiple testing should be applied to all secondary outcomes.
- Reported results (abstract, results, and discussion) should refrain from using casual language (“effect”, “prevent”, “treat”, and “verify”) and be interpreted appropriately. This study design would be considered hypothesis-generating and does not evaluate a causal effect of an intervention.
- Graphical deceptions (with individual data points) of pre and post-measurements are encouraged.
- Age should be included in table 4
- The text on a lack of a control group (lines 299-308) should not be placed in the manuscript as it is. A control group could just receive standard of care and would not have to be fed a “pro-inflammatory diet.” The limitations of this study should be discussed in an objective manner.
Author Response
Consulte el archivo adjunto
